behaviour

noise, marine mammals, mid-frequency sonar, fin whales, behaviour

**Author for correspondence:**
Brandon L. Southall
e-mail: brandon.southall@sea-inc.net

# Behavioural responses of fin whales to military mid-frequency active sonar

Brandon L. Southall[1,2], Ann N. Allen[3,4],
John Calambokidis[3], Caroline Casey[1,2],
Stacy L. DeRuiter[5], Selene Fregosi[1],
Ari S. Friedlaender[1,2], Jeremy A. Goldbogen[6],
Catriona M. Harris[7], Elliott L. Hazen[2,8], Valentin Popov[7]
and Alison K. Stimpert[9]

[1]Southall Environmental Associates, 9099 Soquel Drive, Suite 8, Aptos, CA 95003, USA
[2]Institute of Marine Sciences, Long Marine Laboratory, University of Santa Cruz, 115 McAllister Way, Santa Cruz, CA 95060, USA
[3]Cascadia Research Collective, 218 ½ W 4th Avenue, Olympia, WA 98501, USA
[4]NOAA Pacific Islands Fisheries Science Center, 1845 Wasp Boulevard, Building 176, Honolulu, HI 96818, USA
[5]Calvin University, 3201 Burton Street SE, Grand Rapids, MI 49546, USA
[6]Hopkins Marine Station, Stanford University, 120 Ocean View Boulevard, Pacific Grove, CA 93950, USA
[7]Centre for Research into Ecological and Environmental Modelling, University of St Andrews, The Observatory, St Andrews KY16 9LZ, UK
[8]NOAA Southwest Fisheries Science Center, Environmental Research Division, 99 Pacific Street, Suite 255A, Monterey, CA 93940, USA
[9]Moss Landing Marine Laboratories, San Jose State University, 8272 Moss Landing Road, Moss Landing, CA 95039, USA

BLS, 0000-0002-3863-2068; ANA, 0000-0002-8105-6414;
CC, 0000-0003-3773-2478; ASF, 0000-0002-2822-233X

The effect of active sonars on marine mammal behaviour is a topic of considerable interest and scientific investigation. Some whales, including the largest species (blue whales, *Balaenoptera musculus*), can be impacted by mid-frequency (1–10 kHz) military sonars. Here we apply complementary experimental methods to provide the first experimentally controlled measurements of behavioural responses to military sonar and similar stimuli for a related endangered species, fin whales (*Balaenoptera physalus*). Analytical methods include: (i) principal component analysis paired with generalized additive mixed models; (ii) hidden Markov models; and (iii) structured expert elicitation using response severity metrics. These approaches provide complementary perspectives on the nature of potential changes within and across individuals. Behavioural changes

were detected in five of 15 whales during controlled exposure experiments using mid-frequency active sonar or pseudorandom noise of similar frequency, duration and source and received level. No changes were detected during six control (no noise) sequences. Overall responses were more limited in occurrence, severity and duration than in blue whales and were less dependent upon contextual aspects of exposure and more contingent upon exposure received level. Quantifying the factors influencing marine mammal responses to sonar is critical in assessing and mitigating future impacts.

## 1. Introduction

Sound is critical to marine mammals. The physics of the ocean strongly favours the acoustic channel for communication, predator detection and navigation over most ranges. All marine mammals have well-developed sound production, reception and communication systems that are vital to their fitness and survival [1]. For instance, baleen whales use primarily low frequency sounds (typically less than 1 kHz) for long-range communication, which is important in reproductive and social interactions; toothed whales use active biosonar (typically greater than 10 kHz) to find prey items and orient spatially. It has been widely documented that human noise can interfere with such crucial biological processes through direct and/or indirect auditory, behavioural and physiological effects. Yet we lack a comprehensive understanding of the exposure contexts in which these effects occur and the severity and consequences of these impacts despite substantial scientific and regulatory interest in recent decades [2–7]. Direct measurements of fitness related impacts for endangered species with known noise exposure contexts remain critical. These is also a need for measurements of responses that may influence vital rates (e.g. cessation of foraging) and that can be quantified in energetic terms at the individual level, to inform assessment of population consequences using demographic models (e.g. [8–11]).

Military sonar has received particular attention in ocean policy, management, litigation and research, largely because of lethal mass stranding events of cetaceans around the world associated with tactical mid-frequency active sonar (MFAS) [12]. These stranding events have almost exclusively included odontocete cetaceans (and predominantly several beaked whale species) which generally have strong social organization. However, several events have involved small numbers of baleen whales [13], which typically have less pronounced social order and cohesion. While catastrophic stranding events of baleen whales associated with MFAS are rare, many observational and experimental studies have documented behavioural responses of varying severity in marine mammals, including baleen whales, to various kinds of active sonar (e.g. [4,5,14–20]). Observed responses to sonar in these and other studies include varying degrees of interference with social, vocal, foraging and diving behaviours and more severe behavioural modifications including sustained habitat avoidance. Unlike strandings, these may not directly result in mortality, but such responses may negatively influence vital rates in ways that, depending upon their magnitude and duration, could be detrimental to individuals and populations. Adverse responses could also interact with other stressors (e.g. pollutants) that amplify effects in nonlinear ways that remain poorly understood [7]. A starting point to understanding and modelling population level effects in species for which few data exist is quantifying the type and severity of behavioural responses to a disturbance (e.g. MFAS) in known exposure conditions [11].

The experimental use of controlled exposures of a potential stressor to quantify exposure–response type and probability is a well-established method in biology developed nearly a century ago [21]. When applied to behaviour, this approach measures baseline behaviours in the absence of a stressor and then compares these with behaviours during and following exposure to that stressor. Experimental controls are conducted where all conditions are maintained as during exposure conditions, but sham (e.g. no noise) exposures are used. Technological advances in fine-scale movement and tracking tags allow for field-based methods of conducting controlled exposure experiments (CEEs), which can provide quantitative measurements of key exposure contextual variables (e.g. received sound level, spatial proximity) in combination with detailed information about behavioural state before and after exposure. Recent research has used CEEs to better understand the behavioural responses of various marine mammals to MFAS [4], and more studies have begun to substantially expand sample sizes to include vulnerable whale species (e.g. [5]) which require field studies given their inaccessibility in laboratories. Such data are directly relevant to policy and management planning.

Fin whales are listed as endangered and are federally protected under the US Endangered Species Act of 1973 (16 U.S.C. § 1531 et seq.) and the US Marine Mammal Protection Act of 1972

(16 U.S.C. § 1361 et. seq.). Fin whale population structure is not well understood but there appear to be several populations that inhabit North Pacific waters, and fin whales are present year-round in the Southern California Bight (SCB), particularly in the central and southern areas [22–24]. Fin whale habitat overlaps with the Southern California Range Complex, which is frequently used by the US Navy as the site of MFAS training exercises. Active sonar of some type, including the most powerful MFAS systems, is used on or near a dedicated range facility to the west of Santa Catalina Island, on most days with concentrated periods associated with multiple sources during some periods. Military sonar is less common or concentrated in other parts of the SCB. Fin whales regularly inhabit areas in and around the range and can cover large areas making it somewhat difficult to evaluate the potential for incidental sonar exposure(s). Fin whales in the SCB are at risk of chronic, presumably largely behavioural impacts of repeated sonar exposures that could lead to long-term negative impacts at both the individual and population levels. A single study to date has evaluated fin whale responses to MFAS. Harris *et al*. [25] investigated foraging disruption across three baleen whale species (including a subset of the fin whales included here) and found no significant reduction in fin whale lunge feeding during or following sonar exposure in areas on and around the range facility. To prevent confounding effects, we sought to investigate behavioural responses of fin whales within typical feeding habitats in areas near but not immediately within the most concentrated sonar use areas. We ensured that whales were not exposed to audible incidental sonar (non-experimental) during specified CEE periods, but whales could have possibly been exposed to MFAS or other active sonars in periods prior to their being tagged for this study.

Here we present experimental results quantifying fin whale behavioural responses to MFAS signals like those used in tactical Navy systems. We also presented pseudorandom noise (PRN) signals lacking tonal patterns but having similar frequency, duration, and source levels. As in Tyack *et al*. [16], PRN was used to evaluate whether whales respond similarly to these presumably more novel, less recognizable signals (given the regular use of MFAS in some of these areas) or whether responses to MFAS trigger a distinct response, perhaps an anti-predator response given similarities to some killer whale calls. Stimuli parameters were selected based on MFAS signals these endangered species are known to be exposed to in important feeding areas rather than any similarity to biological signals such as conspecific communication sounds occurring in much different (lower) frequency bands. These experiments were conducted in the same geographical areas, using comparable experimental methods, and the same suite of response type and severity tests as previous studies for another endangered baleen whale species—blue whales (*Balaenoptera musculus*; see [5,26,27]). Combining results from several different analytical methods (principal component analyses (PCAs), generalized additive mixed models (GAMMs), hidden Markov models (HMMs) and expert elicitation of response severity), we assess behavioural changes of fin whales within individuals, between individuals and across multiple behavioural states.

# 2. Methods

## 2.1. Data collection

Data were collected over the summer months from 2010 to 2016 in coastal and offshore areas of the SCB as part of the Southern California Behavioural Response Study (SOCAL-BRS). This multi-year collaborative project investigated the effects of MFAS and other anthropogenic noise on many cetacean species, including both baleen whales and odontocetes. Our experimental and field logistical methods include the same approaches for CEEs as those employed in previous studies with blue whales [5,28]. In brief, a small (approx. 6 m) rigid-hull inflatable boat (RHIB) was used to locate, tag and conduct behavioural focal follows of selected whales. Visual observers collected focal-follow data, which consisted of the position and behaviour of the target whale at each surfacing as well as other information about group type, composition and behaviour. Two archival, animal-borne tag types were used: several different versions of the DTAG [29] and the Acousonde [30]. Both tag types had inbuilt hydrophones (DTAGs sampled at 64, 120 or 240 kHz; Acousondes sampled at 12.6 kHz), as well as pressure sensors and accelerometer and magnetometer sensors (sampled at multiple rates; decimated to either 10 or 25 Hz). Hydrophones provide acoustic data related to exposure levels for experimental stimuli and other sounds in the environment including whale calls and other biological and abiotic signals. Pressure sensors provide information on depth and diving behaviour. Accelerometers and

magnetometers quantify relative three-dimensional movement, orientation and the direction of the earth's magnetic field relative to the whale, which provides measurements of the animal's heading.

Once focal individuals were located and determined to be appropriate candidates for tagging and CEEs (based on research permit requirements and logistical considerations), tagging effort was initiated. After tagging, a minimum 45-min period elapsed before any experimental sequences were conducted to allow for the tagged animal to return to baseline behaviour and minimize any disturbance effects related to the tagging event. The CEE protocol was then initiated with one of three possible experimental treatments determined *a priori* using a randomization procedure (as in [5]). These treatments were: (i) MFAS signals, (ii) PRN signals, and (iii) experimental control (no known noise exposure). The experimental MFAS and PRN signals were identical in spectral and temporal parameters and projected from the same experimental source as described in detail in Southall *et al.* [28].

Experimental signals were projected once every 25 s for the duration of the treatment, simulating common repetition rates in operational military MFAS systems. They were ramped up in 3 dB increments from an initial source level of 160 dB RMS re 1 μPa-m (hereafter dB) to a maximum source level of 210 dB for MFAS signals or 206 dB for PRN signals. The experimental source was suspended to a 10 m transmit depth for all experimental treatments from a central research platform (65′ dive vessel; M/V *Truth*) which was strategically positioned at approximately 1 km from the focal tagged animal. In instances where multiple focal animals were simultaneously tagged and being tracked, some individuals occurred at greater ranges. The CEE sequence consisted of a baseline monitoring period with a specified 30-min pre-exposure phase, followed by a 30-min exposure phase, followed by a 30-min post-exposure period. Tag and focal-follow data collection was maintained in an identical manner across all three CEE phases.

For a subset of CEEs, active acoustic methods were used to measure krill distribution and density in the proximity of whales immediately before and after the three phases of CEE sequences. Detailed methods for the collection and analysis of prey data are provided in [31–34]. After the experimental protocol was conducted and post-playback prey mapping concluded, the focal animal was monitored until the tag detached and was recovered.

For experimental scenarios with playback of either PRN or MFAS, both of which occur in the same 3.5–4.1 kHz frequency band (for additional signal parameters, see [28]) received level (RL) of each individual signal measured on the tag was calculated as the root mean square (RMS) sound pressure level in a one-third octave band centred at 3.7 kHz (as in [16]). A measure of the cumulative sound exposure level (cSEL; in dB re. 1 μPa$^2$-s) was also measured as integrated sound energy across all received sound exposures for each animal (as in [35]).

## 2.2. Data analysis

We strategically applied a suite of analytical methods developed in previous studies with blue whale CEEs to evaluate various response behaviours (e.g. horizontal avoidance, diving, foraging). The objective was to apply established analytical approaches to evaluate the type, probability and magnitude of responses for the same set of whales using several analytical techniques that provide different perspectives at both the group and individual level. Group-level assessments may be more conventionally applied in classic behavioural response studies to understand broad patterns of response. However, analyses of individual responses where exposure conditions are known and can be derived into probabilistic response functions can in some instances more effectively inform applied policy and management decisions. Both group and individual-level analyses have been developed and applied in recent marine mammal behavioural response studies, each with advantages and limitations [4]. The goal here was to strategically apply a suite of approaches used in discrete studies of responses for different subsets of individuals for a closely related species (blue whales) using similar experimental methods. Here, however, for fin whales we sought to apply each of these different analytical approaches to the same subset of individuals.

Specifically, we analysed potential responses across all individuals within several defined behavioural states and specified response categories (see §2.2.1—based on Goldbogen *et al.* [26]) and investigated potential patterns in behavioural state switching (see §2.2.2—based on DeRuiter *et al.* [27]). We also evaluated changes within individuals across the duration of exposure, as well as the timing of any changes that were observed, using a systematic behavioural severity scaling method with expert elicitation (see §2.2.3—based on Southall *et al.* [5]).

## 2.2.1. Principal component analyses and generalized additive mixed models

Following the approach used in Goldbogen *et al.* [26], we used a combination of PCAs and GAMMs to assess the effects of exposure during MFAS, PRN and control CEEs on 12 continuous behavioural metrics. For every dive below 3 m, behavioural metrics were assessed on a dive-by-dive basis and summarized into three categories prior to PCAs. Each category included multiple behavioural metrics calculated as or derived from the original suite of behavioural metrics. Category 1 included dive behaviour metrics (dive time, maximum depth, post dive surface time, descent time, ascent time, bottom time, lunges and breaths). Category 2 included angular (body orientation) metrics (descent pitch, roll, heading; change in descent pitch, roll, heading; ascent pitch, roll, heading; change in ascent pitch, roll, heading). Category 3 included horizontal behaviour metrics (horizontal speed, surface speed, horizontal turning rate, distance to sound source at start and end of dive, change in distance to sound source).

Rather than quantifying response with a single variable, this allowed us to test CEE exposure and contextual effects on a suite of tag and focal-follow derived metrics. The PCA was run using 'princomp' in the *stats* package of the open-source software R (v. 2.15.1). Those PCA eigenvectors with greater than 10% of variance explained were used as response variables in GAMMs. We fitted two GAMMs per eigenvector. The first model assesses the effect of treatment status, specifically pre-exposure, exposure and post-exposure sequences for all treatments:

$$PCA\_Axis \sim f\,(\text{treatment status} \times \text{treatment type} + \text{behavioural state}) + s(\text{Maximum Received Level})$$
$$+ s(\text{Cumulative Sound Exposure}) + s(\text{Average Received Level}).$$

The second model quantifies the response (during exposure only) as a function of CEE exposure type (MFAS, PRN, control):

$$PCA\_Axis \sim f(\text{playback type} \times \text{behavioural state}) + s(\text{Maximum Received Level})$$
$$+ s(\text{Minimum Received Level}) + s(\text{Cumulative Sound Exposure}).$$

Following methods described in detail in Friedlaender *et al.* [32], direct metrics of prey density before and after CEE sequences were incorporated into the dive and angular metric analysis. Since prey data were available for only a subset of the animals ($n = 12$), the paired PCA and GAMM approaches were re-run with the same formulae as above for those deployments with concurrent prey data including acoustically detected bottom depth and prey patch density as response variables in the GAMM. This statistical approach allowed quantitative assessment of: (i) whether behavioural responses occurred (i.e. if treatment status was significant); (ii) whether there was a difference between CEE exposure types (i.e. if playback type was significant); and (iii) whether received level influenced behaviour in addition to or instead of other contextual parameters.

## 2.2.2. Hidden Markov models

We applied HMMs to infer the fin whales' behavioural states from their observed diving behaviour and to evaluate whether consistent behavioural changes occurred across the individuals tested with different treatments. We followed the procedure outlined in DeRuiter *et al.* [27], where details about the methodology can be found.

For each whale, the observable component of the HMM is multivariate such that each observation is a vector of the following five variables calculated for each dive: dive time (in seconds), maximum dive depth (in metres), number of lunges, speed over ground (m s$^{-1}$) and turning angle (in radians). These variables were selected based on previous studies of behavioural responses of baleen whales [5,26] to evaluate potential changes in diving behaviour similar to category 1 metrics described above.

We model the dive time, maximum dive depth and speed over ground with gamma distributions because these variables take positive values and the (empirical) distributions are right skewed. The number of lunges are modelled with a Poisson distribution, and for the turning angles von Mises distribution is used—standard choices for count and angular data, respectively.

For the state-dependent process, we extend the basic HMM approach in two ways. First, to account for differences between tag records, we incorporate discrete-valued random effects in a similar manner to DeRuiter *et al.* [27]. This means that each whale displays one of $K$ different state-switching regimes. $K$ denotes the number of random effects groups or 'contexts', which may relate to the individual's environmental and/or social context. These represent different patterns of switching between the behavioural states, and are intended to account for individual differences between tag recordings.

We allowed the initial state distributions to vary across contexts. We fitted models with the number of contexts $K$ ranging from 1 to 6.

The second extension allows a sound exposure covariate to influence the transition probability matrices (TPMs). The covariates considered are exposure (binary; 1 during exposure and 0 otherwise), exposure phase (categorical; before/during/after exposure), average received playback level during dive (continuous under exposure, 0 when there is no exposure) and maximum received playback level during dive (continuous under exposure, 0 when there is no exposure). We consider two alternative approaches: one where the effect of the covariate is the same across contexts, and one where it is different for each context.

We fitted each model using maximum likelihood. For details on the construction of the likelihood, see DeRuiter *et al.* [27]. Here we note that the log likelihood of the whole dataset was calculated as a sum of the likelihoods of the individual time series assuming independence between them. Models were fitted in R statistical software using customized code.

Based on prior knowledge of fin whale behaviour as well as a preliminary analysis focused on baseline models without random effects or sound exposure covariates (see electronic supplementary material, figure S1), we considered HMMs with three states [36]. We then used Akaike's information criterion (AIC) to select the number of contexts, $K$, to choose the best of the candidate covariates, and to decide whether its effect was the same across contexts. For the final model, we also obtained Wald confidence intervals for exposure covariate coefficients using the inverse Fisher information matrix.

In the optimization procedure for the likelihood, the choice of initial values is crucial. To avoid reaching a local rather than a global maximum, we fitted the baseline model with 2000 randomly chosen sets of starting values. When fitting the more advanced HMMs with random effects, for the starting values of the parameters of state-dependent distributions, we used the estimates from the baseline model and chose the remaining ones at random. The convergence was fast. Nevertheless, for the best model we tried 3400 different starting values. In addition, we jittered the parameter estimates (excluding those of the state-dependent distributions) for the starting values in 1000 further trials to ensure there was not a global maximum in the vicinity of the one that we found.

### 2.2.3. Structured expert elicitation and behavioural response scoring

Southall *et al.* [5] applied expert evaluation using an established 10-point behavioural response severity scale to assess behavioural responses of blue whales to simulated and operational Navy MFAS. Further, Southall *et al.* [5] used Mahalanobis distance methods of quantifying responses for the same blue whale subjects in parallel with expert evaluation. Here, in parallel with the quantitative evaluative metrics of behavioural response identified above, we applied and adapted well-established, structured qualitative assessment methods for evaluating changes in behaviour. This approach was founded on expert evaluation methods used in other ecological applications and involved a structured process in which subject-matter experts assess identical graphical and quantitative representations of behaviour during baseline and exposure conditions. Six subject-matter experts, which included some individuals who participated in field experiments, conducted the scoring. This included three individuals each in two discrete groups. The groups met simultaneously but were separately asked to identify whether and when responses of various types occur and to identify their severity according to specified criteria. An adjudicator that was intimately familiar with the study and scoring process served to answer clarifying questions during the independent group scoring.

Here we applied identical methods to those used for expert evaluation in Southall *et al.* [5], with one notable exception. Rather than evaluate them separately, we calculated Mahalanobis distance metrics for fin whales (using the same suite of behaviours as in Southall *et al.* [5]) and provided those time-series results in addition to dive profile, lunge rate, minimum specific acceleration (MSA), heading variance, and horizontal speed to the teams of expert evaluators. An annotated map noting the whale track, colour coded for exposure phase and the track of the sound source vessel were also provided to expert evaluators for each tagged animal (see electronic supplementary material, Information; figures S2–S4). Scorers were blind to individual whale ID, date and location of CEEs, exposure treatment type (MFAS, PRN, control), precise timing of exposure signals, and exposure RL. Each CEE was presented in randomized order in terms of the date that the experiment was conducted. The evaluation teams used identical severity scale metrics and instructions as in Southall *et al.* [5]. Results from each of two groups of scorers were adjudicated between the two groups by the independent mediator who was not involved in the original scoring process to a single assessment of response type, severity score (0 for no response; 9 for most severe response on a 10-point scale), and time of occurrence.

Exposure–response probability functions were then generated using recurrent event survival analyses to assess time-to-event changes using marginal stratified Cox proportional hazards models fitted to the severity score data [37]. In this modelling framework, results from individual CEEs were used to estimate the likelihood of response as a function of exposure received level (in cSEL) and contextual covariates. Models were fitted to two broad categories of response severity: low severity (severity score 1–3) and moderate severity (severity score 4–6). There were no instances of high severity scores (score 7–9) as evaluated by expert scoring, so this category of severity was not included. For each CEE, the time of first occurrence of each response severity level was included in the model. For CEEs with a severity score of 0 (no response), the cSEL for the entire exposure sequence was used and the data were labelled as right censored, meaning that no response was detected up to this RL.

We fitted models to data from all CEEs and included source-animal range (m) at the start of the exposure phase, signal type (MFAS or PRN) and behavioural state in the pre-exposure period (deep feeding, shallow feeding, non-feeding) as covariates. Observations were assumed to be correlated within individuals but independent between individuals, given that none of the experiments were conducted with individuals in coherent paired groups. The standard errors of the model estimates were corrected for the correlations within individuals using a grouped jackknife procedure [38]. All possible model combinations from the null model through to a full interaction model were fitted, and AIC-based model selection was used to select the model with best fit. For the selected model, we tested that the proportional hazards assumption was met [37]. Analyses were conducted in R statistical software version 3.6.3 using the Survival package [39], and exposure–response functions were generated as survival curves from the fitted models using the survfit function in that package.

# 3. Results

A total of 21 fin whales were tagged between 2010 and 2016; 19 animals were instrumented with DTAGs, and two with Acousondes. Tag deployment durations were typical for these suction-cup attached archival tags, ranging from 4 to 19 h and enabling sufficient baseline and CEE data according to the experimental design specified. There were nine deep feeding, seven shallow feeding and five non-feeding animals. A total of 11 MFAS, 4 PRN and 6 controls were conducted on the tagged animals. Prey mapping was conducted during 12 of the experimental scenarios (table 1).

## 3.1. PCA/GAMM results

PCA results for whale dive behaviour and angular movement are summarized in table 2. Dive time, depth and number of lunges accounted for 85% of the variance in category 1 (dive behaviour metrics). For the primary axis of angular metrics, descent pitch, total roll, max pitch, ascent pitch and start heading explained 95% of the variance. Our paired PCA–GAMM analyses revealed significant differences related to sound exposure. The data provide strong evidence of an effect of sound exposure on the primary component axis (eigenvector) for two out of three parameters (dive metrics, $p < 0.05$ for Max and Average RL; $r^2$ adj = 0.485; orientation metrics, $p < 0.05$ for Max and Average RL; $r^2$ adj = 0.491; horizontal metrics, $p > 0.05$; $r^2$ adj = 0.16).

Although there was variation in individual's responses (thus significance is not tested here), bar plots of the non-dimensional responses indicate that fin whale behavioural responses during CEEs were not affected by either behavioural state or noise exposure type (figure 1).

Rather, we found received level was the primary factor driving response probability for both dive and angular metrics (figure 2).

For those animals with prey data, dive metrics accounted for 95% of the total variance in the PCA. We found that the ratio of prey patch depth to dive metrics highlighted a large change in behaviour relative to changes in prey patch depth from before to during MFA playbacks. The GAMMs showed a similar result where increasing bottom depths led to increased likelihood of response, concurrent with treatment type having a significant effect for the angular metrics but not for the dive metrics.

## 3.2. Hidden Markov model results

The model selected based on the lowest AIC value among 11 competing models was a 3-state HMM with five contexts and the maximum received level (MaxRL) as a covariate whose effect was the same across contexts. Although HMM states are a modelling construct and do not necessarily correspond to the kinds

**Table 1.** Fin whale tags, CEEs conducted, and exposure contexts. Asterisks indicate animals with prey mapping either pre- or post-CEE or both.

| behavioural state at CEE onset | CEE type | subject identification | CEE date | CEE number | start times for CEE phases (local - PDT) | | |
|---|---|---|---|---|---|---|---|
| | | | | | pre-exposure | exposure (min) | post-exposure |
| deep-feeding | CONTROL (n = 1) | 20160914_B020-BP | 16 Sep 2016 | 2016_04 | 1323 | 1353 (30) | 1423 |
| deep-feeding | MFAS (n = 5) | bp10_236a | 24 Aug 2010 | 2010_03 | 1120 | 1150 (30) | 1220 |
| | | bp10_239a | 27 Aug 2010 | 2010_05 | 1204 | 1234 (30) | 1304 |
| | | bp13_216a* | 4 Aug 2013 | 2013_11 | 1539 | 1609 (19) | 1628 |
| | | bp16_256a* | 12 Sep 2016 | 2016_03 | 1512 | 1542 (30) | 1612 |
| | | 20160912_B014-BP | 12 Sep 2016 | 2016_03 | 1512 | 1542 (30) | 1612 |
| deep-feeding | PRN (n = 3) | bp10_245a | 2 Sep 2010 | 2010_11 | 1322 | 1352 (30) | 1422 |
| | | bp10_247a | 4 Sep 2010 | 2010_13 | 1314 | 1344 (30) | 1414 |
| | | bp12_217a* | 4 Aug 2012 | 2012_03 | 1422 | 1452 (30) | 1522 |
| shallow-feeding | CONTROL (n = 4) | bp13_258a* | 15 Sep 2013 | 2013_15 | 1252 | 1322 (30) | 1352 |
| | | bp13_258b* | 15 Sep 2013 | 2013_15 | 1252 | 1322 (30) | 1352 |
| | | bp13_258c | 15 Sep 2013 | 2013_15 | 1252 | 1322 (30) | 1352 |
| | | bp13_265a | 22 Sep 2013 | 2013_19 | 1312 | 1342 (30) | 1412 |
| shallow-feeding | MFAS (n = 3) | bp13_257b* | 14 Sep 2013 | 2013_14 | 1530 | 1600 (30) | 1630 |
| | | bp13_259a* | 16 Sep 2013 | 2013_16 | 1046 | 1116 (30) | 1146 |
| | | bp15_236a* | 24 Aug 2015 | 2015_06 | 1357 | 1427 (30) | 1457 |
| non-feeding | CONTROL (n = 1) | bp14_259a* | 16 Sep 2014 | 2014_08 | 1322 | 1352 (30) | 1422 |
| non-feeding | MFAS (n = 3) | bp10_236b | 24 Aug 2010 | 2010_03 | 1120 | 1150 (30) | 1220 |
| | | bp13_139a | 19 May 2013 | 2013_01 | 0944 | 1014 (30) | 1044 |
| | | bp15_229a* | 17 Aug 2015 | 2015_02 | 1211 | 1241 (30) | 1311 |
| non-feeding | PRN (n = 1) | bp12_294a* | 20 Oct 2012 | 2012_06 | 1416 | 1446 (30) | 1516 |

**Table 2.** Loadings for whale dive behaviour and angular movement.

| loadings dive behaviour: | Comp.1 | Comp.2 | Comp.3 | | |
|---|---|---|---|---|---|
| dive time | −0.570 | −0.674 | 0.470 | | |
| max depth | −0.598 | | −0.800 | | |
| lunges | −0.564 | 0.737 | 0.373 | | |
| **loadings angular metrics:** | **Comp.1** | **Comp.2** | **Comp.3** | **Comp.4** | **Comp.5** |
| descent pitch | 0.491 | 0.131 | −0.491 | 0.701 | |
| sum roll | −0.537 | | 0.206 | 0.609 | 0.543 |
| max pitch | −0.542 | −0.160 | | 0.237 | −0.786 |
| ascent pitch | −0.391 | | −0.831 | −0.287 | 0.258 |
| start heading | 0.155 | 0.973 | 0.134 | | −0.105 |

of biological behavioural states assigned from field and tag observations (table 1) of the animals [40–42], the six fitted state-dependent distributions (figure 3) illustrate that here the HMM states do correspond to biologically interpretable behavioural states.

Broadly speaking, state 1 may be characterized as deep feeding, state 2 as non-feeding (which include shallow dives) and state 3 as shallow feeding. Of the six coefficients that quantify the effect of MaxRL on

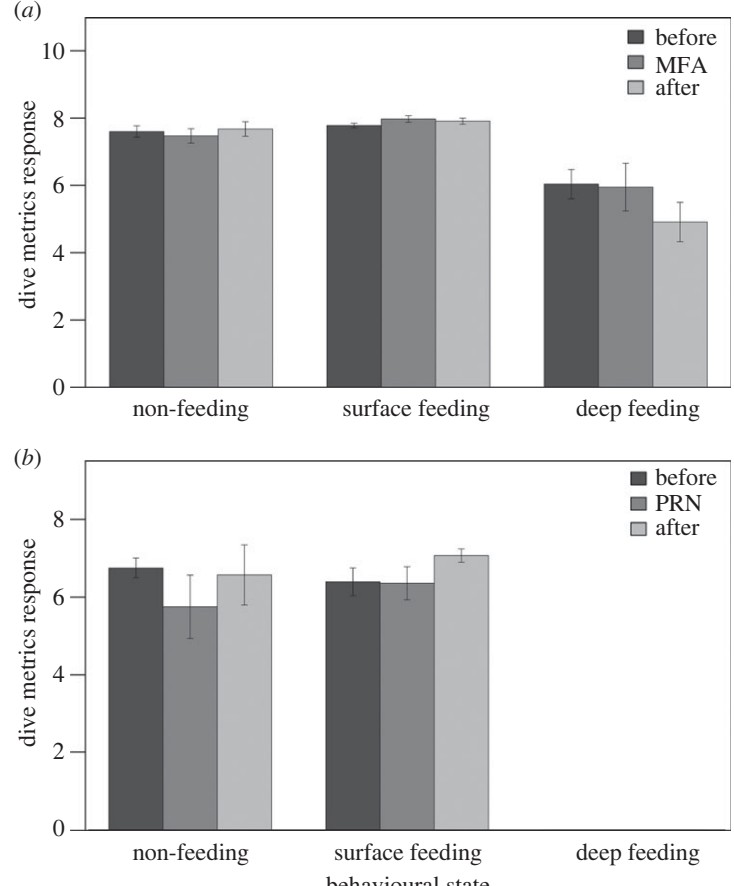

**Figure 1.** Bar plots for diving response metrics during CEE phases for MFA (*a*) and PRN (*b*) experiments across all behavioural states sampled. Bars represent standard error.

state transition probabilities, only the one affecting the transition from shallow non-feeding (state 2) to deep feeding (state 1) had a 95% confidence interval that excluded 0. All else remaining constant, an increase in MaxRL thus increases the odds of switching to a deep-feeding state (state 1) compared to other states.

The numeric values of the estimated coefficients are difficult to interpret directly. Instead, we visualize (figure 4) the effects of the covariate by plotting TPMs for each context in two scenarios: when MaxRL is zero (no exposure) and when MaxRL is fixed to its mean value during exposure (130 dB cSEL).

For some of the contexts (1, 3 and 4; figure 4 from left to right), one or two states had high persistence, while others were occupied rarely and briefly. This result is not surprising given the relatively short tag durations (less than 24 h). The effect of exposure is particularly prominent for contexts 1 and 5, where a typical value of MaxRL substantially increases the probability of switching from the shallow non-feeding state to the deep diving state. These findings are strengthened for the same model by the results of local decoding of the states under discrete random effects (see [27] for details). An example time series of one whale's observed dive data plus the decoded states and the timing of the exposure are provided below (figure 5). This illustrates that the start of the exposure often coincides with a switch from the shallow non-feeding state to the deep feeding state, which is also associated with an increased number of lunges.

## 3.3. Expert scoring and event survival results

Expert evaluation of tag and focal-follow time-series data identified behavioural changes in five of the 21 fin whale CEEs, ranging in severity score from 2 to 6 (table 3). Exposure RL, changes identified, and severity and confidence level assessed are provided for each whale.

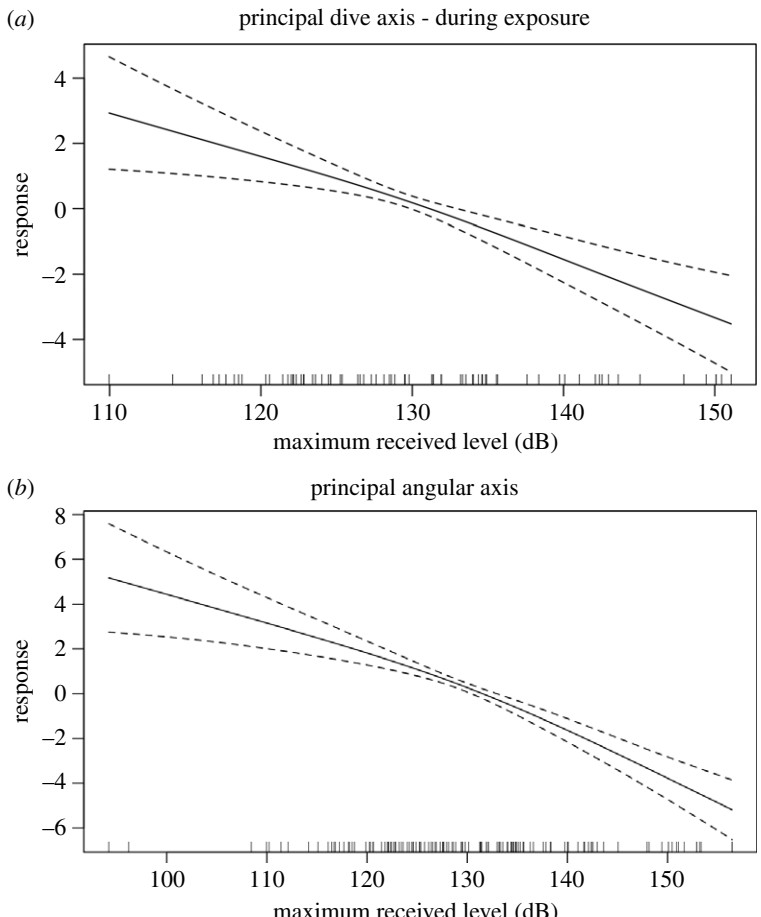

**Figure 2.** Partial response plots for (*a*) horizontal diving metrics (derived from dive time, max depth, feeding lunges) and (*b*) angular movement metrics (derived from aspects of pitch, roll and heading) shown as a function of maximum received levels (RLs) across all behavioural states for MFA and PRN CEEs. We found an increase in (i) dive times and (ii) greater max pitch, greater descent pitch, and lower ascent pitch with an increase in RL.

All whales identified as initiating behavioural response(s) as a function of exposure were from either MFAS or PRN CEEs (i.e. no whales responded in control CEEs), and all three behavioural states had at least one responding whale. Notably, both expert groups observed that whale bp15_229a exhibited a change in behaviour at the onset of MFAS exposure, with an increase in lunges. A similar scenario occurred in blue whales [5] where whales began feeding during exposure. Because the severity scale developed by Southall *et al*. [3] and modified by Miller *et al*. [17] did not include the onset of feeding as a change (as they focused on presumed adverse changes), this animal was not scored as exhibiting a negative/adverse behavioural response. However, it was noted, as it is here, that a behavioural state change (to foraging) likely occurred.

The Cox proportional hazards model that was selected based on the lowest AIC scores that also met the proportional hazards assumption (global *p*-value from $\chi^2$-test = 0.186) only included signal type as a covariate, with no significant difference between the two signal types (*p* = 0.219). Models that included source-animal range and behaviour state as covariates had lower AIC values but violated the proportional hazards assumption, so were not considered further. The ΔAIC between the model which included signal type as a covariate and the null model with no covariates was less than 0.5, providing weak support for inclusion of signal type. The predicted exposure–response probability functions for the two different response severity levels (low severity: 1–3; moderate severity: 4–6) for both signal types show a higher probability of response with higher signal level for both signal types, with no significant difference in probability of response between low and moderate severity scores. Exposure–response prediction plots suggest a higher probability of responding to PRN signals than MFAS signals. The difference, however, is not significant as is evident from the wide, overlapping confidence intervals (figure 6). No plots are given for control CEEs as no whales were determined through expert scoring to exhibit behavioural responses of any severity.

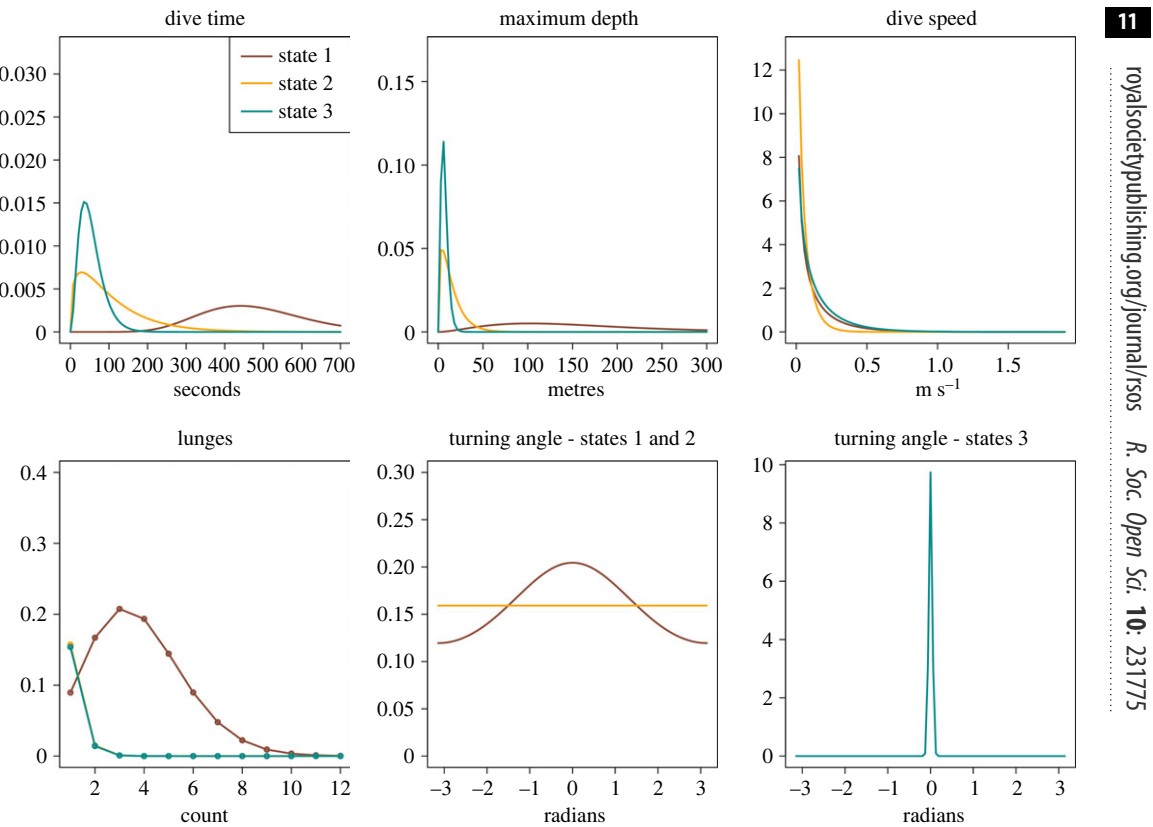

**Figure 3.** Fitted state-dependent distributions for the six behavioural data-streams in each of three states (state 1, 'deep feeding'; state 2, 'non-feeding'; state 3, 'shallow feeding'), according to the HMM.

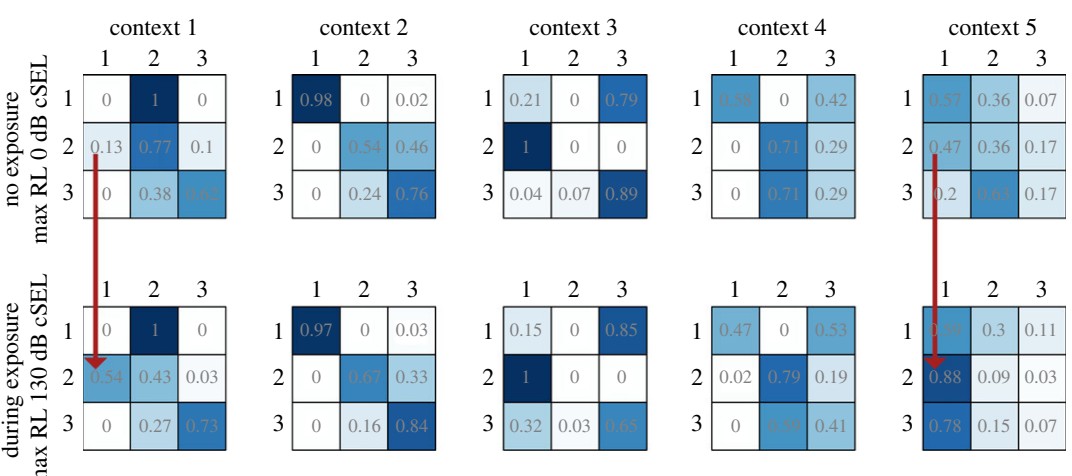

**Figure 4.** Transition probability matrices under different exposure conditions. The five identified contexts represent different patterns of switching between the behavioural states recordings (1, deep feeding; 2, non-feeding; 3, shallow feeding), and account for individual differences between tag recordings. Darker blue shading indicates higher probability of transition. The upper row of panels shows the five contexts when MaxRL is zero (no exposure) and lower row shows the same five contexts when MaxRL is set to its mean value during exposure. Red arrows highlight two contexts where the increased probability of switching from the non-feeding to the deep-feeding state during noise exposure is large enough to have practical importance.

## 4. Discussion

This study provides the first experimentally controlled measurements of fin whale responses to active military sonar and similar noise stimuli. Here we adapted and synthesized three previously applied analytical methods [5,26–28,32] to evaluate potential behavioural responses in known and controlled

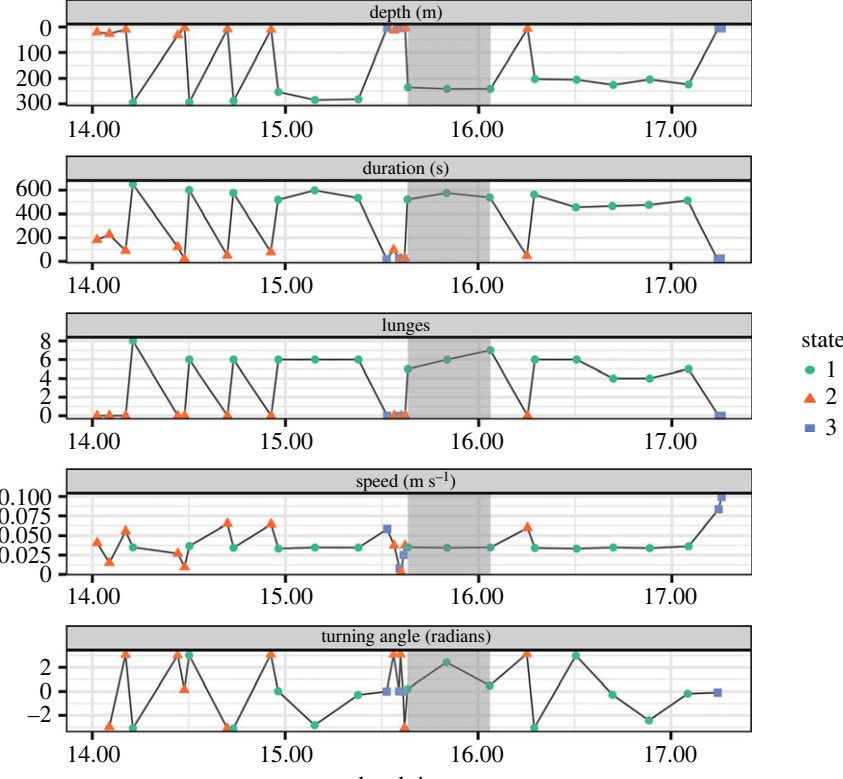

**Figure 5.** Time series showing data input to the HMM and a 'local decoding' for states defined in the 3-state HMM for an example whale (bp13_258c). The CEE exposure phase is shaded grey.

noise exposure conditions. We used these complementary methods to evaluate the evaluate the roles of key exposure contextual variables (e.g. exposure presence, type, proximity and received level; environmental factors) in driving the occurrence, type, and magnitude of behavioural responses of fin whales. It is important to note that a smaller sample size was obtained for fin whales in this study relative to previous work on blue whales (21 versus 52, respectively), which may account for some of the differences observed between species. We present the combined results of the strategically complementary analytical methods applied to the entire fin whale dataset (all individuals included in each analysis) here. This includes both across individual assessments (PCA/GAMM, HMMs) to evaluate broader response patterns as well as within individual time-series analyses (expert evaluation with responses aggregated into probabilistic risk functions). Each approach provides differing degrees of temporal and spatial resolution, with combined results suggesting a different mode of response to sonar compared to blue whales evaluated with similar analytical methods applied over multiple studies. We consider here the nature of response for fin whales observed with these complementary methods in relation to blue whale responses. We note, however, the contextual similarities for CEEs conducted here with fin whales and blue whales, both in terms of experimental and in terms of the potential for prior exposure to incidental actual MFAS given the proximity of experimental locations to areas of higher sonar concentration.

We found that a subset of fin whales exposed to mid-frequency sonar and similar frequency band and duration stimuli clearly exhibited behavioural responses of mild to moderate severity. These responses had notable differences from those of blue whales. Fin whale responses were generally less common overall for similar exposures (33% response of any type detected in expert elicitation in fin whales versus 63% overall response probability in blue whales). Response occurrence and type were less dependent upon behavioural state at the time of exposure for fin whales than blue whales. Conversely, fin whale responses that did occur were more related to received exposure levels than was observed in blue whales. As with blue whales, fin whales were likely to return to baseline conditions relatively quickly after noise exposure ended (within post-exposure phases in most instances). Also, as was observed in blue whales [5], some fin whales initiated deep diving and or/feeding states during noise exposure. This may simply reflect a lack of a response to the noise stimuli for animals that have located a new foraging patch than an explicit response to noise *per se*. However, neither this nor

**Table 3.** Expert scoring results for fin whales in MFAS, PRN and no-noise control conditions across all behavioural states.

| behavioural state at CEE onset | CEE type | subject identification | CEE date | prey mapping? | | expert scoring | | | |
|---|---|---|---|---|---|---|---|---|---|
| | | | | pre-CEE | post-CEE | cSEL at change point or max (dB re 1 $\mu Pa^2$ s) | change? (confidence) | severity score | change description |
| deep-feeding | CONTROL ($n=1$) | 20160914_B020-BP | 16 Sep 2016 | no | no | — | no (med) | — | — |
| deep-feeding | MFAS ($n=5$) | bp10_236a | 24 Aug 2010 | no | no | 160 | no (high) | — | — |
| | | bp10_239a | 27 Aug 2010 | no | no | 145 | yes (med) | 2 | brief changes in respiration rate; minor change in dive profile |
| | | bp13_216a | 4 Aug 2013 | yes | yes | 153 | no (med) | — | — |
| | | bp16_256a | 12 Sep 2016 | yes | yes | 162 | no (med) | — | — |
| | | 20160912_B014-BP | 12 Sep 2016 | no | no | 164 | no (med) | — | — |
| deep-feeding | PRN ($n=3$) | bp10_245a | 2 Sep 2010 | no | no | 166 | no (med) | — | — |
| | | bp10_247a | 4 Sep 2010 | no | no | 111 | yes (high) | 6 | moderate cessation in feeding; minor avoidance of source (5) |
| | | bp12_217a | 4 Aug 2012 | yes | yes | 153 | no (high) | — | — |
| shallow-feeding | CONTROL ($n=4$) | bp13_258a | 15 Sep 2013 | yes | yes | — | no (med) | — | — |
| | | bp13_258b | 15 Sep 2013 | yes | yes | — | no (high) | — | — |
| | | bp13_258c | 15 Sep 2013 | yes | yes | — | no (low) | — | — |
| | | bp13_265a | 22 Sep 2013 | no | no | — | no (med) | — | no change identified according to specified criteria but groups noted onset of foraging during CEE |

**Table 3.** (*Continued.*)

| behavioural state at CEE onset | CEE type | subject identification | CEE date | prey mapping? | | expert scoring | | | |
|---|---|---|---|---|---|---|---|---|---|
| | | | | pre-CEE | post-CEE | cSEL at change point or max (dB re 1 $\mu$Pa$^2$ s) | change? (confidence) | severity score | change description |
| shallow-feeding | MFAS (*n* = 3) | bp13_257b | 14 Sep 2013 | yes | no | 105 | yes (low) | 4 | minor avoidance of source |
| | | bp13_259a | 16 Sep 2013 | yes | yes | 156 | no (med) | — | — |
| | | bp15_236a | 24 Aug 2015 | yes | yes | 156 | no (high) | — | — |
| non-feeding | CONTROL (*n* = 1) | bp14_259a | 16 Sep 2014 | yes | yes | — | no (med) | — | — |
| non-feeding | MFAS (*n* = 3) | bp10_236b | 24 Aug 2010 | no | no | 129 | yes (med) | 4 | minor avoidance of source |
| | | bp13_139a | 19 May 2013 | no | no | 149 | no (med) | — | — |
| | | bp15_229a | 17 Aug 2015 | yes | no | 151 | no (high) | — | — |
| non-feeding | PRN (*n* = 1) | bp12_294a | 20 Oct 2012 | yes | yes | 101 | yes (high) | — | moderate change in diving behaviour |

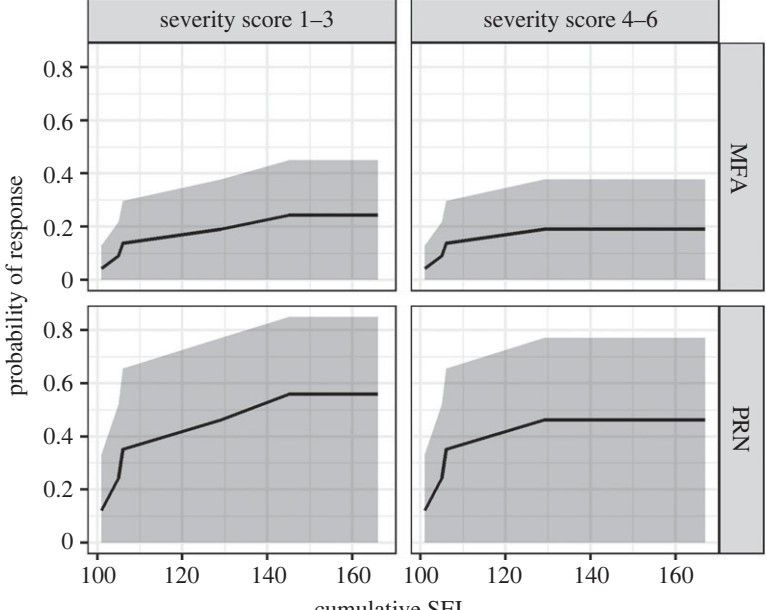

**Figure 6.** Fin whale exposure–response probability functions for MFA and PRN CEEs for low (1–3) and moderate (4–6) severity scores. Black lines show predicted response probability as a function of received cSEL (dB re 1 $\mu Pa^2$ s) and grey areas indicate 95% confidence intervals.

whether there was any potential for experimental stimuli to affect prey behaviour can be definitively evaluated at this point and deserves further investigation.

The PCA/GAMM analyses revealed several clear conclusions for fin whale CEEs. Specifically, the large majority of variability in the selected dive and angular metrics used was accounted for in three parameters each: dive time, maximum depth, and number of lunges; and positive pitch, negative roll, and negative heading. This result suggests that fin whales in this study were less variable overall in their diving and orientation behaviour than blue whales that had more complex PCA results described by Goldbogen *et al.* [26] and Friedlaender *et al.* [32]. Further, unlike blue whale responses, we found no significant influence of behavioural state or noise exposure type (MFAS or PRN) on fin whale response. We did, however, find both maxRL and average RL to be a significant factor driving fin whale response. Higher RLs were associated with deeper and longer dives, more lunges, higher ascent pitch, greater roll and greater start heading. Finally, our findings suggest that foraging success was not compromised in exposed fin whales. This contrasts the results presented in Friedlaender *et al.* [32] that found blue whale foraging was negatively impacted by their behavioural response to sonar. Although samples where prey data were successfully collected before and following CEE exposure periods were more limited, our results suggest that fin whales that did respond by avoiding sound sources sought out other patches in the vicinity rather than simply ceasing feeding.

The results from the HMM analyses provide several similar conclusions regarding whale behaviour and the nature of responses that did occur in fin whale CEEs. Across individuals and CEE types, fin whales were less responsive to noise exposure than blue whales studied by DeRuiter *et al.* [27]. Interestingly, the only state change probability identified as a function of exposure level (maxRL) was an increase in the probability of non-feeding whales switching to a deep diving state with a higher expected number of lunges. This similarly suggests that fin whales that do respond to the presence of noise during CEEs by changing diving behaviour are more likely to dive deeper, longer, and in one case initiate feeding. This could be explained by animals avoiding the immediate CEE area and location (near the surface) and feeding in suitable surrounding areas, effectively representing a mild response with likely few energetic consequences (assuming foraging is as successful as in areas avoided).

Finally, expert elicitation of responses to evaluate exposure occurrence and severity within individuals provided some additional explicit details on the nature and duration of exposures that did occur. Of perhaps greatest note is the limited overall number of identified responses. None occurred during no noise control sequences, and low-to-moderate severity changes were identified in just five of 15 total whales exposed to noise. These exposures were conducted using identical methods, similar exposure levels, similar geographical and spatial contexts, and at similar times of year; results were

analysed using comparable analytical approaches with almost all the same experts contributing to the severity scoring assessment as with blue whales [5]. A smaller sample size of fin whales than blue whales may explain some of the wider confidence intervals around the exposure–response functions and the low probability of response for MFAS exposures. The expert elicitation results also suggest an overall lower probability of response in fin whales than in blue whales and milder and/or briefer types of changes relating to important behaviours such as feeding. There was a possible indication of differential and stronger response to PRN relative to MFAS exposures, which could perhaps be related to the relative novelty of such signals within an environment in which Navy operational MFAS signals are common. However, we were unable to fully resolve how significant this possible differential response may have been due to the limited number of CEEs conducted.

Next steps and needed progressions include more directly exploring the relationship between geographical range from sources to receivers and the associated received level. This will allow us to explore whether animals may be 'ranging' in mediating response probability (as suggested by and discussed in Southall *et al.* [5] for blue whales). Additional remaining issues include gaining a better understanding of responses to actual, operational sources (some of which have already been collected). Ongoing studies are also increasing the duration of data before and after CEEs to better quantify baseline behaviour and the duration of responses that do occur. Finally, the apparently greater response to novel stimuli used here (PRN) relative to presumably more familiar signals in this area of frequent MFAS usage, as seen previously in blue whales, suggests that where large-scale introduction of novel signal types are likely to occur in the future for the Navy or other offshore industrial operators, targeted studies ahead of their introduction or strategic monitoring when they first occur should be conducted to inform assessments of impact or potential mitigation as needed.

This study provides important relevant information about the nature and probability of behavioural response in an endangered baleen whale species that is commonly exposed to high-power military sonar in this and other geographical areas. This study adds to the collective knowledge of responses of baleen whales to military sonar, providing direct information to improve and inform environmental impact assessments and effective mitigation.

**Ethics.** All research activities for this study were authorized and conducted under US National Marine Fisheries Service permit no. 14534; Channel Islands National Marine Sanctuary permit no. 2010-004; US Department of Defense Bureau of Medicine and Surgery (BUMED) authorization; a federal consistency determination by the California Coastal Commission; and numerous institutional animal care and use committee (IACUC) authorizations.

**Data accessibility.** Raw data and further HMM analyses are available at: https://people.ucsc.edu/~elhazen/SOCAL/SOCAL_CEE_FinWhales.html.

Supplementary material is available online [43].

**Declaration of AI use.** We have not used AI-assisted technologies in creating this article.

**Authors' contributions.** B.L.S.: conceptualization, data curation, formal analysis, funding acquisition, investigation, methodology, project administration, supervision, validation, visualization, writing—original draft, writing—review and editing; A.N.A. and C.C.: data curation, formal analysis, investigation, visualization, writing—original draft, writing—review and editing; J.C.: conceptualization, funding acquisition, investigation, methodology, project administration, resources, supervision, writing—review and editing; S.L.D.: formal analysis, investigation, software, validation, visualization, writing—original draft, writing—review and editing; S.F.: data curation, formal analysis, validation, visualization, writing—original draft, writing—review and editing; A.S.F.: conceptualization, formal analysis, investigation, supervision, visualization, writing—original draft, writing—review and editing; J.A.G.: conceptualization, formal analysis, investigation, project administration, supervision, visualization, writing—original draft, writing—review and editing; C.M.H.: formal analysis, visualization, writing—original draft, writing—review and editing; E.L.H.: data curation, formal analysis, investigation, software, validation, visualization, writing—original draft, writing—review and editing; V.P.: formal analysis, methodology, software, validation, visualization, writing—original draft, writing—review and editing; A.K.S.: formal analysis, investigation, software, visualization, writing—original draft, writing—review and editing.

All authors gave final approval for publication and agreed to be held accountable for the work performed therein.

**Conflict of interest declaration.** We declare we have no competing interests.

**Funding.** Primary funding for the SOCAL-BRS project was initially provided by the US Navy's Chief of Naval Operations Environmental Readiness Division and subsequently by the US Navy's Living Marine Resources (LMR) Program. Additional support for environmental sampling and logistics was also provided by the Office of Naval Research, Marine Mammal Program.

**Acknowledgements.** This study could not have been conducted without the tireless support of many dedicated field personnel. We acknowledge the contributions of: Kristin Southall, Peter Tyack, Jay Barlow, Shannon Rankin, Fleur Visser, Annie Douglas, Greg Schorr, Erin Falcone, Katy Laveck, Jeff Foster, Todd Pusser, Glenn Gailey, Doug Nowacek, Louise Burt, John Hildebrand, Ron Morrissey, Greg Juselis, Jolie Harrison, Tami Adams, Sarah Wilkin, Ned Cyr, Frank Stone, Bob Gisiner and Mike Weise.

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
