## [Peer Review File · Royal Society Open Science]

Review History

RSOS-230162.R0 (Original submission)

Review form: Reviewer 1

Is the manuscript scientifically sound in its present form?

Yes

Are the interpretations and conclusions justified by the results?

Yes

Is the language acceptable?

Yes

Do you have any ethical concerns with this paper?

No

Have you any concerns about statistical analyses in this paper?

No

Recommendation?

Accept with minor revision (please list in comments)

Comments to the Author(s)

Comments in the attached file (see Appendix A).

Decision letter (RSOS-230162.R0)

Dear Dr Southall

On behalf of the Editors, we are pleased to inform you that your Manuscript RSOS-230162 "Behavioral responses of fin whales to military mid-frequency active sonar" has been accepted for publication in Royal Society Open Science subject to minor revision in accordance with the referees' reports. Where applicable, please find the referees' comments along with any feedback from the Editors below my signature.

Please submit your revised manuscript and required files (see below) no later than 7 days from today's (ie 10-Jul-2023) date. Note: the ScholarOne system will 'lock' if submission of the revision is attempted 7 or more days after the deadline. If you do not think you will be able to meet this deadline please contact the editorial office immediately.

Kind regards,

Royal Society Open Science Editorial Office
Royal Society Open Science
openscience@royalsociety.org

on behalf of Dr Denise Greig (Associate Editor) and Kevin Padian (Subject Editor)
openscience@royalsociety.org

Associate Editor Comments to Author (Dr Denise Greig):

Thank you for this interesting submission. I like the reviewer's ideas for more making the paper more accessible to readers who have not read all the preceding CEE papers. In addition, I would like to see more description in the figure captions so that they can be more easily understood without going back through the text. Thanks again, I look forward to the revision.

Specific questions:

line 233. needs a closed parenthesis

line 304. missing a period

line 411-12. Could these definitions of the State 1, 2, and 3 please be added to the Figure 3 caption?

Could the 5 contexts also be described in the Figure captions for Figures 3, 4 and 5?

I am having trouble interpreting Figure 4 - could you please explain what the arrows are showing in the caption and what a darker color blue signifies? To me it looks like an increase in the non-feeding state during exposure so I am clearing missing something.

line 458-459. The shift to feeding during exposure is interesting...do you have any data on the impact of the sound sources on prey species behavior? As an extreme example, I am thinking about some of the early pile driving efforts before bubble curtains were used: fish died and floated to the surface when the pile driving started and birds flocked to the site to feed.

Line 495. I think the word "while" can be deleted.

Reviewer comments to Author:

Reviewer: 1

Comments to the Author(s)

Comments in the attached file (2023-0327_RSOS-230162_Southall-etal_RX_Finners_MFA_COMMENTS-Bowles.pdf)

Author's Response to Decision Letter for (RSOS-230162.R0)

See Appendix B.

RSOS-230162.R1 (Revision submission)

Review form: Reviewer 1

Is the manuscript scientifically sound in its present form?

Yes

Are the interpretations and conclusions justified by the results?

Yes

Is the language acceptable?

Yes

Do you have any ethical concerns with this paper?

No

Have you any concerns about statistical analyses in this paper?

No

Recommendation?

Accept

Comments to the Author(s)

The authors have done a good job of addressing my earlier comments. The paper should be accepted.

One technical comment:

Line 165 – Briefly explain the utility of the magnetometer in quantifying the whales' behavior.

I also found a few editorial/typographical issues in case they're useful:

Line 331 vs. 340-345. Lines 340 to about 345 would improve logical flow if introduced before the material starting in line 331.

Line 533-534 “here” appears twice, remove after “complementary methods”?

Figures:

P 35, L 32-39

Typo - “the each of three states”

P 36. Please indicate where the reader can find a description of the 5 contexts.

Decision letter (RSOS-230162.R0)

Dear Dr Southall

On behalf of the Editors, we are pleased to inform you that your Manuscript RSOS-230162.R1 "Behavioral responses of fin whales to military mid-frequency active sonar" has been accepted for publication in Royal Society Open Science subject to minor revision in accordance with the referees' reports. Please find the referees' comments along with any feedback from the Editors below my signature.

Please submit your revised manuscript and required files (see below) no later than 7 days from today's (ie 16-Aug-2023) date. Note: the ScholarOne system will 'lock' if submission of the revision is attempted 7 or more days after the deadline. If you do not think you will be able to meet this deadline please contact the editorial office immediately.

Kind regards,
Royal Society Open Science Editorial Office

on behalf of Dr Denise Greig (Associate Editor) and Kevin Padian (Subject Editor)
 openscience@royalsociety.org

Associate Editor Comments to Author (Dr Denise Greig):

The revision answers all of my questions and the figure captions are much clearer (thank you!). This is a nice addition to the extensive and important body of work investigating the behavior responses of cetaceans to anthropogenic noise.

Minor notes:

In the introduction where you first mention the PRN signals (lines 133-135), it might be worth explaining their role in the experiment. It wasn't clear to me until the discussion where you talk about novel sounds as opposed to the MFAS that they are fairly routinely exposed to.

Like the other reviewer, I would love a description of the 5 contexts (line 292) - is that all in DeRuiter 2017?

Line 545....delete "and" so that it reads "...fin whale responses that did occur were more related to received exposure levels..."

Line 548...add closed parenthesis after "instances".

Reviewer comments to Author:

Reviewer: 1

Comments to the Author(s)

The authors have done a good job of addressing my earlier comments. The paper should be accepted.

One technical comment:

Line 165 - Briefly explain the utility of the magnetometer in quantifying the whales' behavior.

I also found a few editorial/typographical issues in case they're useful:

Line 331 vs. 340-345. Lines 340 to about 345 would improve logical flow if introduced before the material starting in line 331.

Line 533-534 "here" appears twice, remove after "complementary methods"?

Figures:

P 35, L 32-39

Typo - "the each of three states"

P 36. Please indicate where the reader can find a description of the 5 contexts.

===PREPARING YOUR MANUSCRIPT===

one version should clearly identify all the changes that have been made (for instance, in coloured highlight, in bold text, or tracked changes);

a 'clean' version of the new manuscript that incorporates the changes made, but does not highlight them. This version will be used for typesetting.
Please ensure that any equations included in the paper are editable text and not embedded images.

===PREPARING YOUR REVISION IN SCHOLARONE===

- If you are requesting a discretionary waiver for the article processing charge, the waiver form must be included at this step.
- If you are providing image files for potential cover images, please upload these at this step, and inform the editorial office you have done so. You must hold the copyright to any image provided.
- A copy of your point-by-point response to referees and Editors. This will expedite the preparation of your proof.

- Ensure that your data access statement meets the requirements at <https://royalsociety.org/journals/authors/author-guidelines/#data>. You should ensure that you cite the dataset in your reference list. If you have deposited data etc in the Dryad repository, please only include the 'For publication' link at this stage. You should remove the 'For review' link.
- If you are requesting an article processing charge waiver, you must select the relevant waiver option (if requesting a discretionary waiver, the form should have been uploaded, see 'File upload' above).
- If you have uploaded any electronic supplementary (ESM) files, please ensure you follow the guidance at <https://royalsociety.org/journals/authors/author-guidelines/#supplementary-material> to include a suitable title and informative caption. An example of appropriate titling and captioning may be found at https://figshare.com/articles/Table_S2_from_Is_there_a_trade-off_between_peak_performance_and_performance_breadth_across_temperatures_for_aerobic_scope_in_teleost_fishes_/3843624.

RSOS-231775.R0 (Revision)

Decision letter (RSOS-231775.R0)

Dear Dr Southall:

I am pleased to inform you that your manuscript entitled "Behavioral responses of fin whales to military mid-frequency active sonar" is now accepted for publication in Royal Society Open Science.

Please remember to make any data sets or code libraries 'live' prior to publication, and update any links as needed when you receive a proof to check - for instance, from a private 'for review' URL to a publicly accessible 'for publication' URL. It is also good practice to add data sets, code and other digital materials to your reference list.

Royal Society Open Science is a fully open access journal. A payment may be due before your article is published. Please note that, if the corresponding author of your paper is based at an institution covered by one of our Transformative Agreement deals, your fees may be covered by the deal – please check the list of eligible institutions

at <https://royalsociety.org/journals/authors/read-and-publish/read-publish-agreements/>. The Royal Society has partnered with Copyright Clearance Center's (CCC's) RightsLink service to allow authors to pay article processing charges or page charges. After your manuscript has been accepted, the corresponding author will receive an email from CCC with the subject "Please submit your article processing/open access charge(s)/page charges" inviting you to pay your charges or request an invoice. The email from CCC will come from the email domain @copyright.com (if you have any queries regarding fees, please see <https://royalsocietypublishing.org/rsos/charges> or contact authorfees@royalsociety.org). If you request an invoice, it will be sent to you from CCC.

It is important to be cautious about payment scams. If you receive an email or text message requesting payment and have any concerns, we recommend contacting us through our website, rather than clicking on any links. The Royal Society will never ask you to make a direct payment.

Follow Royal Society Publishing on Twitter: [@RSocPublishing](https://twitter.com/RSocPublishing)
Follow Royal Society Publishing on Facebook:
<https://www.facebook.com/RoyalSocietyPublishing/>
Read Royal Society Publishing's blog:
<https://royalsociety.org/blog/blogsearchpage/?category=Publishing>

Appendix A

Please give an indication of the ongoing military sonar activity, such as annual number of exposures, intervals between, duration of exposures at any given time, etc., to make it possible to compare the experiments with ongoing exposures, to the extent known. There are several places in the introduction and discussion where this information would be helpful (e.g., 113-115, 551-553). The question is also pertinent when explaining the experimental exposures – how much did they resemble ongoing exposure and in what way?

Based on the results and discussion together, I suggest an editing pass beginning either in the introduction or when introducing the analyses to 1) give the big-picture rationale for the different analysis types (why is expert response scoring needed, for instance?), and 2) to add an ending summary for the results in the form of a table by whale/experiment with a summary of the multiple analysis outcomes for that whale and a description interpreting the outcome. This paper follows a great deal of methodological development in previous papers and a table would help make the outcomes clear without having to dig back into the previous literature. For example, 532 is the first time we are told that they had direct evidence from only one whale that it initiated feeding at depth (does not show up in Table 3). As another example, the sentence in 522-523 is more difficult to understand without a list of what every whale did based on all the information available.

Please give an indication of the change in RL that was produced by diving to depth during a response. Would diving to depth have been more effective than surfacing? What made the authors relate these dives to feeding?

In the discussion: please address the usefulness of the analyses (and the ensemble of results) now that their techniques have been applied to a new species. For example, how did expert solicitation complement measures based on data from their subject whales? Was there any parameter or component of the tag & focal track data that could have served as an appropriate proxy for the expert scores?

Specific Comments:

27-29: Two uses of the word timely in this sentence. Remove the first?

79: I'm a little uncomfortable with "sub-lethal" as a term for behavioral responses with the potential to affect vital rates like reproductive success – it seems to have become a de facto standard, but less understandable for effects that don't involve physical injury. On the other hand, I don't like the alternatives used in the past and the term is explained well in this paragraph. May be more helpful if defined explicitly before using it.

113 – Širović

188: Maybe make the point that this band encompasses the MFAS/PRN signals. Is there any evidence that fin whales use this band for particular functions (e.g., communication?).

343-344: Were any of the whales traveling in groups? The explanation here begs the question of between-whale independence.

394-395: Reword – the PCA relationship between change in prey patch depth and MFA is not clear. The GAMM results are clear.

427: Tag duration by subject would be helpful –incorporate into Table 3?

497: Unexpected end of sentence here – should be a comma?

503-505: Reword – the meaning is unclear in this sentence and the point they are making is important. Should “later” be “latter”? Do they mean that an animal that has found a new foraging patch may be less likely to respond? 522-523 a clear explanation, so this sentence should set the stage.

512: A place where the distinction between “that” and “which” is important. If they mean “which”, please precede by a comma.

533: When the whales dove, would it have affected the RL substantially? How would it have differed from a return to the surface?

554: This hypothesis is very helpful, but also emphasizes the utility of adding detail about ongoing exposures in the introduction.

Fig 2 – the legend and the plot titles don’t seem to match. Perhaps explain the plot titles more fully?

Fig 4 – Specify what the red arrows indicate.

A couple of issues found in the references:

620: The DTIC document doesn’t come up – try <https://apps.dtic.mil/sti/citations/ADA601387>

630: The journal lists this as a 2019 paper

Appendix B

Behavioral responses of fin whales to military mid-frequency active sonar Responses to Reviewer Comments

We were very happy to receive the decision to accept this article for publication in RSOS with minor revision. We appreciate the useful comments and suggestions from both Reviewer 1 and from the Associate Editor (Dr. Greig). We considered each of them and address how they were addressed directly in order below in preparation of our revised manuscript.

Reviewer 1 Comments

Reviewer 1 General Comments:

- Please give an indication of the ongoing military sonar activity, such as annual number of exposures, intervals between, duration of exposures at any given time, etc., to make it possible to compare the experiments with ongoing exposures, to the extent known. There are several places in the introduction and discussion where this information would be helpful (e.g., 113-115, 551-553). The question is also pertinent when explaining the experimental exposures – how much did they resemble ongoing exposure and in what way?

Response: Good point and substantial additional introductory and discussion for context was added. It's challenging for a number of logistical and security reasons to be extremely specific on this and it could easily be an analysis unto it's own. But this is quite relevant and the fact that there is some sonar use in the area but these experiments were for the most part not in the middle of the greatest concentration on the SOAR range is definitely relevant. The similarity to ongoing exposures is a bit of a separate question and was also addressed to some extent but was expanded in both the intro and the discussion.

- Based on the results and discussion together, I suggest an editing pass beginning either in the introduction or when introducing the analyses to 1) give the big-picture rationale for the different analysis types (why is expert response scoring needed, for instance?), and 2) to add an ending summary for the results in the form of a table by whale/experiment with a summary of the multiple analysis outcomes for that whale and a description interpreting the outcome. This paper follows a great deal of methodological development in previous papers and a table would help make the outcomes clear without having to dig back into the previous literature. For example, 532 is the first time we are told that they had direct evidence from only one whale that it initiated feeding at depth (does not show up in Table 3). As another example, the sentence in 522-523 is more difficult to understand without a list of what every whale did based on all the information available.

Response: We had tried to do #1 in the last paragraph of the introduction and first paragraph of the methods section related to the analyses, but can see the point that this may have not been sufficiently explicit. Additional rationale and explanation there for the reasons to look at changes within and between animals using established methods applied to other species were

added. Given the nature of the analyses, where results are pooled across individuals and behavioral states for two of the three analyses, it's really only possible or meaningful to identify changes within each of the individuals for the expert scoring assessment. We clarify this expand the discussion of and made some clarifications and expansions to Table 3, rather than adding another table that would really only include those results again if aiming to list responses within each individual.

- Please give an indication of the change in RL that was produced by diving to depth during a response. Would diving to depth have been more effective than surfacing? What made the authors relate these dives to feeding?

Response: See response to specific question on this below. From both RL modeling and measured RLs on tags, there is not a clear pattern in terms of reduction or RL with depth. In some instances in fact as show in table 3 and in the supplementary materials, animals diving deeper had higher RLs. Given this pattern and the extent of additional detail, space, and figures that would be required to really get into it, we chose to simplify any discussion or speculation on vertical avoidance. We did add some clarifying text and in the figure captions to show the presence of lunges in actual data relating to states of shallow or deep feeding.

- In the discussion: please address the usefulness of the analyses (and the ensemble of results) now that their techniques have been applied to a new species. For example, how did expert solicitation complement measures based on data from their subject whales? Was there any parameter or component of the tag & focal track data that could have served as an appropriate proxy for the expert scores?

Response: Some sections added here to try and return to the useful suggestion for additions in the methods section as to why these different/complementary methods were applied to the same set of whales.

Reviewer 1 Specific Comments:

27-29: Two uses of the word timely in this sentence. Remove the first?

Response: First one removed.

79: I'm a little uncomfortable with "sub-lethal" as a term for behavioral responses with the potential to affect vital rates like reproductive success – it seems to have become a de facto standard, but less understandable for effects that don't involve physical injury. On the other hand, I don't like the alternatives used in the past and the term is explained well in this paragraph. May be more helpful if defined explicitly before using it.

Response: This makes sense and we think it works better without the term sub-lethal at all. Removed in both places and just left the discussion about responses that do not result in direct mortality as it is defined.

113 – Širović

Response: Corrected

188: Maybe make the point that this band encompasses the MFAS/PRN signals. Is there any evidence that fin whales use this band for particular functions (e.g., communication?).

Response: Clarification and a reference for additional signal parameters given here. Didn't seem the right place to make the point about this band in relation to fin whale bioacoustics but a short consideration of this added in the introduction – anything in this band is likely audible to fin whales and might be used in recognizing other species or environmental sounds but is generally not part of their primary communication.

343-344: Were any of the whales traveling in groups? The explanation here begs the question of between-whale independence.

Response: Added a clarification that none of the CEEs were conducted with lead-trail coherent pairs. Some of the experiments did involve multiple individuals in what could be considered broader feeding aggregations, but all were separated by several or many km and none specifically were paired.

394-395: Reword – the PCA relationship between change in prey patch depth and MFA is not clear. The GAMM results are clear.

Response: We did not make any changes here as we believe the wording accurately captures the results.

427: Tag duration by subject would be helpful –incorporate into Table 3?

Response: There was relatively little variability across whales in tag duration and all were less than a day. We added a short explanation and a range in the results section, but would rather not make Table 3 more complicated as there is already a lot going on.

497: Unexpected end of sentence here – should be a comma?

Response: Reworded this into two simpler sentences drawing specific contrasts between blue and fin whales.

503-505: Reword – the meaning is unclear in this sentence and the point they are making is important. Should “later” be “latter”? Do they mean that an animal that has found a new foraging patch may be less likely to respond?

Response: Clarifying changes in both sentences made accordingly.

512: A place where the distinction between “that” and “which” is important. If they mean “which”, please precede by a comma.

Response: “...that...” is clearer - changed

522-523 a clear explanation, so this sentence should set the stage.

Response: We think this was just an observation as this sentence is intended to set the stage for this paragraph.

533: When the whales dove, would it have affected the RL substantially? How would it have differed from a return to the surface?

Response: We simplified this by removing the speculation on vertical avoidance which was that and just added ‘location (near the surface)’ in the preceding sentence, which doesn’t really lose much. In many instances, we actually see higher RLs at depth in the upper few hundred meters due to downward refracting conditions but don’t see that opening that up without a complex analysis that would include quite a lot more explanation and not change any of the conclusions is needed.

554: This hypothesis is very helpful, but also emphasizes the utility of adding detail about ongoing exposures in the introduction.

Response: Good point – as described above some additional context on MFAS and PRN signal types added in the introduction.

Fig 2 – the legend and the plot titles don't seem to match. Perhaps explain the plot titles more fully?

Response: Yes mismatch between the two – propose changing figure panels to mirror the caption which keeps parallel order with Table 2. Corrected that and added some additional explanation about the underlying parameters in figure 2 caption (tying it in with Table 2).

Fig 4 – Specify what the red arrows indicate.

Response: Added to figure caption and per other suggestions and from the associate editor added additional clarification.

- A couple of issues found in the references:

620: The DTIC document doesn't come up – try <https://apps.dtic.mil/sti/citations/ADA601387>

630: The journal lists this as a 2019 paper

Response: Corrections made for first one. The second is one of the online was 2019; formal printed is 2020. We changed to 2019 as what does come up with the journal but it's referenced elsewhere (and in Google Scholar as 2020). The doi gets you to the reference and checked.

Associate Editor General Comments:

Thank you for this interesting submission. I like the reviewer's ideas for more making the paper more accessible to readers who have not read all the preceding CEE papers. In addition, I would like to see more description in the figure captions so that they can be more easily understood without going back through the text. Thanks again, I look forward to the revision.

Response: Substantial additions made in trying to accomplish this without adding too much additional text as already near/over expected limit. But addressed these and especially in clearer, more explicit figure captions.

Associate Editor Specific questions:

line 233. needs a closed parenthesis

Response: Added

line 304. missing a period

Response: Added

line 411-12. Could these definitions of the State 1, 2, and 3 please be added to the Figure 3 caption?

Response: Good idea - done

- Could the 5 contexts also be described in the Figure captions for Figures 3, 4 and 5?

Response: They're not really relevant in Figure 3 as these are the underlying dive parameters used in the model selection process to define the 3-state HMM. But we aimed to clarify each of these figure captions and specify what the contexts mean, which are essentially different patterns in state switching and/or persistence.

- I am having trouble interpreting Figure 4 - could you please explain what the arrows are showing in the caption and what a darker color blue signifies? To me it looks like an increase in the non-feeding state during exposure so I am clearing missing something.

Response: Substantial modifications in the figure caption to try and explain further, paralleling descriptions in the text.

- line 458-459. The shift to feeding during exposure is interesting...do you have any data on the impact of the sound sources on prey species behavior? As an extreme example, I am thinking about some of the early pile driving efforts before bubble curtains were used: fish died and floated to the surface when the pile driving started and birds flocked to the site to feed.

Response: From what we know about the invertebrate prey they forage on (which isn't much) they would not be expected to hear or more likely feel sounds at these high frequencies. It opens kind of a speculative can of worms and there is little known about sound reception in any inverts and none in krill. But we added a simple statement that for these species and what little is known it is unlikely but has not been investigated.

- Line 495. I think the word "while" can be deleted.

Response: Removed and simplified sentence.